# Free Sugars Intake, Sources and Determinants of High Consumption among Australian 2-Year-Olds in the SMILE Cohort

**DOI:** 10.3390/nu11010161

**Published:** 2019-01-13

**Authors:** Gemma Devenish, Rebecca Golley, Aqif Mukhtar, Andrea Begley, Diep Ha, Loc Do, Jane A. Scott

**Affiliations:** 1School of Public Health, Curtin University, Perth 6102, Australia; gemma.devenish@curtin.edu.au (G.D.); Aqif.Mukhtar@curtin.edu.au (A.M.); a.begley@curtin.edu.au (A.B.); 2College of Nursing and Health Sciences, Flinders University, Adelaide 5000, Australia; rebecca.golley@flinders.edu.au; 3Australian Research Centre for Population Oral Health, The University of Adelaide, Adelaide 5000, Australia; diep.ha@adelaide.edu.au (D.H.); loc.do@adelaide.edu.au (L.D.)

**Keywords:** free sugars, food frequency questionnaire, 24-h recall, 2-day food record, dietary intakes, food sources, sociodemographic determinants, early childhood, complementary feeding

## Abstract

In the first 2 years of life, it is important to limit exposure to foods high in free sugars, in order to lay foundations for lifelong eating patterns associated with a reduced risk of chronic disease. Intake data at this age is limited, so compliance with recommendations is not known. This analysis describes free sugars intakes, food sources and determinants of high consumption among Australian children at 2 years of age. Free sugars intakes were estimated using a customized Food Frequency Questionnaire, and median usual free sugars intake at 2 years was 22.5 (Interquartile Range (IQR) 12.8–37.7) g/day, contributing a median 8% of the estimated energy requirement (EER). Based on the EER, most children (71.1%) exceeded the World Health Organization recommendation that <5% of energy should come from free sugars, with 38% of participants exceeding the <10% recommendation. Children from households with the greatest socioeconomic disadvantage were more likely to exceed the 10% recommendation (Prevalence Ratio (PR) 1.44, 95% Confidence Interval (95% CI) 1.13–1.84), and be in the top tertile for free sugars intake (PR 1.58, 95% CI 1.19–2.10) than the least disadvantaged. Main sources of free sugars were non-core foods, such as fruit juice, biscuits, cakes, desserts and confectionery; with yogurt and non-dairy milk alternatives the two notable exceptions. Improved efforts to reduce free sugars are needed from the introduction of solid food, with a particular focus on fruit juice and non-core foods.

## 1. Introduction

The first 2 years of life is a critical time for establishing food preferences and eating behaviors that lay the foundations for long-term dietary habits [1,2]. It is important that during this time children are exposed to a wide variety of nutritious foods, with limited intakes of foods higher in saturated fat, added salt and added sugars [3,4]. The recommendations of the World Health Organization (WHO) to limit free sugars to less than 10%, and ideally less than 5% of energy intake [5], are particularly relevant during the early years while taste preferences are being established [6,7]. The WHO defines free sugars as “monosaccharides and disaccharides added to foods and beverages by the manufacturer, cook or consumer, and sugars naturally present in honey, syrups, fruit juices and fruit juice concentrates” (p. 4) [5]. Consumption of free sugars above these recommendations is associated with increased body weight and dental caries from early childhood through to adulthood [5]. When introducing solids it is important to limit exposure to foods and drinks high in free sugars, in order to establish lifelong healthy eating patterns [4,7].

The Australian National Nutrition and Physical Activity Survey (NNPAS) reported an increasing trend in free sugars intakes from 2–3 years of age throughout childhood, peaking in adolescence and then declining gradually through adulthood [8]. This was the first national survey to report free sugars intakes, supported by an updated Australian nutrient database (AUSNUT2011–13), which includes values for added and free sugars for the first time [9]. Just over 50% of children aged 2–3 years in the NNPAS exceeded the WHO recommendation that less than 10% of total energy should come from free sugars, and 93% exceeded the less than 5% recommendation [8]. Unsurprisingly, most free sugars came from non-core foods and beverages, accounting for 81% of free sugars intakes across all ages [8].

At present, national nutrition monitoring does not collect data for children less than 2 years of age, so knowledge of dietary intakes as children transition to the family diet is limited. Recent Australian studies around this age raise concerns around early exposure to non-core foods [10,11,12,13], inadequate fruit and vegetable consumption [14], inadequate iron intake [15,16,17,18], and poor overall compliance with dietary guidelines [19]. Findings also suggest that dietary behaviors track over time, emphasizing the importance of establishing healthy food patterns from an early age [11,12,13,19]. Some of these studies have investigated determinants, and report associations between socioeconomic factors and diet quality [14,19,20,21]. None of these studies report free sugars intakes.

In the first few months of life, intakes of free sugars are likely to be minimal, as breastmilk and the majority of animal milk-based infant formula do not contain free sugars. At present, little is known about when and in what forms free sugars enter the diet, as children move through the weaning period towards a dietary pattern reflective of their family’s diet. Free sugars intakes at 1 year of age have recently been reported for participants in the Study of Mothers’ and Infants’ Life Events affecting oral health (SMILE) [22]. This research extends this work to report findings for the same cohort at 2 years of age. 

The primary aims of this study are to describe free sugars intakes and food sources among Australian children at 2 years of age, and investigate sociodemographic determinants of high free sugars consumption. Secondary aims are to determine compliance at 2 years with the WHO guideline for sugars intake for adults and children [5]; and to examine tracking of free sugars intake between 1 and 2 years of age.

## 2. Materials and Methods 

### 2.1. Recruitment and Data Source

Data came from the Study of Mothers’ and Infants’ Life Events affecting oral health (SMILE). This birth cohort consisted of 2147 mothers and 2181 children, including 34 sets of twins, recruited from the major maternity hospitals in Adelaide from July 2013 to August 2014. Study methods, including sampling, recruitment and data collection, are described in detail elsewhere [23]. All new mothers with sufficient English competency and not intending to move out of the greater Adelaide area within a year were invited to participate. Women from hospitals in lower socioeconomic areas were oversampled to account for higher attrition rates, which resulted in a cohort that was generally representative of the socioeconomic profile reported by the Pregnancy Outcome unit for South Australian births in 2013 [24]. Data were collected at baseline and when the child was three, six, twelve and twenty-four months of age. A range of dietary, dental and sociodemographic risk factors were captured, via the parents’ choice of online, paper or telephone questionnaire [23]. The Southern Adelaide Clinical Human Research Ethics Committee approved the study (HREC/50.13, approval date: 28 Feb 2013) as did the South Australian Women and Children Health Network (HREC/13/WCHN/69, approval date: 7 August 2013).

### 2.2. Dietary Intake Data

Dietary data used in this analysis were collected using a customized 89-item, semi-quantitative Food Frequency Questionnaire (SMILE-FFQ) emailed or posted to parents when their child reached 2 years of age. At this time they were also invited to book their child in for a dental examination with the SMILE team. In addition to the dental assessment at this appointment, the child’s weight and standing height were measured by a trained member of the SMILE team, using calibrated equipment and following standardized methodology [25].

The SMILE-FFQ was designed to estimate usual intake of total and free sugars in Australian toddlers and was validated against repeat 24-h recalls in an external cohort [26]. A matching database customized for the SMILE-FFQ was linked to responses using Microsoft Access version 15 (Microsoft Corporation, 2013, Washington, DC, USA) to generate an estimated usual intake of total and free sugars. Free sugars values for each line item in the FFQ were determined by adapting the method by Louie, et al. [27], to incorporate the WHO definition of free sugars [5].

As the SMILE-FFQ was not designed to capture total energy intake, it was not possible to determine the percentage of energy intake from free sugars at 2 years of age. As an alternative, Estimated Energy Requirement (EER) was determined from age and weight data, and used to calculate percentage of EER from free sugars. The method for calculating EER used in the Australian Nutrient Reference Values [28] was followed, using an equation for children aged 13–35 months that was developed by the Food and Nutrition Board of the Institute of Medicine [29]. Participants with no weight measurement who returned a complete FFQ (*n* = 125) were assigned a reference EER using the Nutrient Reference Value for age and gender [28]. Sensitivity analysis was conducted by repeating the primary analyses with these participants removed.

For the food group analysis, each line item in the SMILE-FFQ was linked to a food group from the AUSNUT2011–13 database, mostly at the sub-major food group level. The groups ‘Fruit products and dishes’ and ‘Snack foods’ were linked at the major food group level only, as the line items in the FFQ related to these broader categories. Conversely, the AUSNUT2011–13 sub-major group ‘Infant formula and foods’ was too broad, so these items were further broken down into minor food groups. Out of the 132 sub-major food groups in the AUSNUT2011–13 database; 55 were linked to at least one item in the FFQ. Those that were not represented were either foods not commonly consumed by pre-school aged children (e.g., alcohol and coffee) or foods low in sugars (e.g., meat, eggs, vegetables etc.).

Dietary collection methods at 1 year, along with findings for free sugars intakes, food groups and determinants are reported in detail elsewhere [22]. Briefly, after the child’s first birthday, 3 non-consecutive days of dietary data were obtained via a 24-h recall and 2-day food diary, following standardized methods and protocols [30,31] that were tailored to this age. Data were entered into FoodWorks version 8 (Xyris Software, 2012–2017, Brisbane, Australia) and analyzed with the current Australian food composition database, AUSNUT2011–13 [32], supplemented with additional infant and toddler food items that were lacking in the national database. The Multiple Source Method [33] was applied to the resulting three days of dietary data to obtain the usual daily intake of total sugars, free sugars and energy for each child, and the percentage of energy from free sugars was calculated.

### 2.3. Sociodemographic Data

Sociodemographic data collected via self-report questionnaire at recruitment were used to generate the following categorical explanatory variables for the determinant analysis: Mother’s age at baseline (‘<25’, ‘25–34’ and ‘≥35’ years); Mother’s pre-pregnancy weight status using Body Mass Index (BMI) [34] (‘<25’, ‘25–29.99’ and ‘≥30’ kg/m^2^); Mother’s country of birth (‘Australia and New Zealand’ and ‘Other’); Mother’s education (‘high school/vocational’, and ‘some university and above’); total number of children (‘1’, ‘2’ and ‘≥3’) and infant sex (‘male’ and ‘female’). Postcode was used to derive a measure of socioeconomic status using the Index of Relative Socioeconomic Advantage and Disadvantage (IRSAD), which was grouped into five decile pairs [35].

### 2.4. Statistical Methods

Participants with nutrient intakes ≥ 3 standard deviations from the mean for free sugars at either time point, or energy intakes at 1 year, were considered extreme misreporters and excluded from that time point [18]. This excluded a total of 25 participants from the 1 year (*n* = 11), 2 year (*n* = 13) or both (*n* = 1) datasets. Characteristics of participants with plausible and complete dietary data and those without were compared using the Pearson Chi Square test for categorical variables and the independent samples *t*-test for continuous variables. 

Nutrient intake distributions were right-skewed so nonparametric methods (median, interquartile range) were used to describe nutrient intakes and percentage of EER from free sugars. The percentage of children with free sugars intakes ≥5% and ≥10% of EER are also reported. 

For the food group contributions at 2 years, the mean free sugars (g) for each food group were calculated for all participants and consumers only, at the major and sub-major food group levels, and the minor food group level for infant foods. Consumers were defined as those who reported any valid response other than “never or rarely” in the SMILE-FFQ, for at least one line item within each food group.

Due to the different dietary assessment methods at the two time points, statistical comparisons of free sugars intakes at 1 and 2 years focused on movement relative to the cohort, rather than absolute values. Participants were ranked into tertiles of free sugars consumption at 1 and 2 years of age, and movement between tertiles was compared using crosstabulations and the Weighted Kappa test. SPSS version 25 was used for these analyses (IBM SPSS Statistics for Windows, New York, NY, USA), with statistical significance set at *p* < 0.05.

Sociodemographic factors identified in the literature as being associated with poor diet quality in early childhood were investigated as determinants of high free sugars intakes with generalized regression models, using a log-binomial link function in SAS PROC GENMOD (SAS Institute Inc., Cary, NC, USA). The outcome variables were the highest free sugars tertile at 2 years, and free sugars intakes ≥ 10% of EER at 2 years. The models estimated adjusted prevalence ratios (PRs) and associated 95% confidence intervals (95% CI). Participants with data missing for one or more of the explanatory variables (*n* = 105 10%) were excluded from this analysis (Figure 1). An interaction between mother’s country of birth, child’s sex and free sugars intake observed in the SMILE cohort at 1 year of age [22] was investigated with the 2 year dietary data. However, preliminary analysis revealed that this interaction was not significant at two years, and so was not included in the models for this study.

## 3. Results

### 3.1. Participants

Figure 1 depicts participant flow for the analysis datasets. There were 1043 participants with plausible and complete 2 year dietary data that were included in descriptive and food group analysis. Table 1 compares characteristics of these participants to non-responders. Plausible and complete 1 year dietary data were available for 816 participants, of whom 682 also provided data at 2 years for comparison. Participant characteristics of this sub-set are presented in Appendix A.

### 3.2. Free Sugars Intakes at 2 Years of Age

Median free sugars intake at 2 years of age was 22.5 (IQR 12.8–37.7) grams per day, providing a median 8% of EER (Table 2). Mean intake was 29.3 (SD 23.7) grams per day, providing a mean 10.4% (SD 8.6%) of EER. For most (71.1%) of the children in this cohort, free sugars intake exceeded 5% of their EER, with 38.4% of children exceeding 10% of EER.

### 3.3. Food Group Contribution to Free Sugars Intakes

Table 3 provides a breakdown of food group contribution to free sugars intake by major and sub-major food groups among all participants and consumers only. At the major food group level, cereal-based products and dishes (19% of free sugars), non-alcoholic beverages (16%), milk products and dishes (16%), infant formula and foods (15%), sugar products and dishes (12%) and confectionery and cereal nut/fruit/seed bars (9%) were key contributors to free sugars. 

Drilling down to the sub-major food groups, the greatest overall contribution to free sugars came from fruit and vegetable juices and drinks (11%), followed by infant custards and yogurts (10%) and yogurts that were not specifically identified for infants (9%). The next highest contributors were cakes, muffins, doughnuts and cake-type desserts (8%), and sweet biscuits (7%), followed by sugar, honey and syrups (5%) and chocolate based confectionery (5%). 

Unflavored dairy milk substitutes contributed just over 4% of free sugars to the whole cohort, but was one of the highest contributors to free sugars among those 24% of participants who consumed these products. Mean free sugars from this group among consumers was 5.2 g, similar to fruit and vegetable juices and drinks at 5.5 g among consumers (57% of the cohort). These were only secondary to infant and toddler formula, which provided 7.7 g of free sugars to consumers (14% of the cohort). By comparison, dairy milks do not contain any free sugars, nor does breastmilk.

### 3.4. Changes in Free Sugars Consumption from 1 to 2 Years of Age

Table 4 compares tertiles of free sugars between 1 and 2 years of age. Nearly half the cohort (45%) remained in the same tertile from 1 to 2 years of age, with 16% of participants categorized in the highest tertile at both time points. Among those who provided both 1 and 2 year data, there was fair agreement between usual free sugars intake relative to the cohort at the two time points (Weighted Kappa 0.241, *p* < 0.001) [36].

### 3.5. Determinants of High Free Sugars Intakes at 2 Years of Age

Table 5 reports factors associated with high consumption of free sugars at 2 years. In total, 938 participants with complete 2 year dietary data and data for all of the explanatory variables were included in the final multivariable model. Maternal age, country of birth, parity, and level of socioeconomic disadvantage were all independently associated with being in the top tertile of free sugars intake and of having free sugars intakes ≥ 10% EER. There was no association observed between both free sugars outcomes and mother’s pre-pregnancy BMI or child’s sex.

Children of mothers born outside of Australia and New Zealand were more likely to be in the top tertile for both free sugars intake (PR 1.58, 95% CI 1.28–1.94) and free sugars ≥ 10% of EER (PR 1.59, 95% CI 1.33–1.90). Children with two or more siblings were also more likely to be in the top tertile (PR 1.52, 95% CI 1.17–1.97) and to exceed 10% of EER from free sugars (PR 1.39, 95% CI 1.11–1.76) than those with no siblings. Children born to mothers who were less than 25 years old were also more likely to be in the top tertile (PR 1.50, 95% CI 1.07–2.12) and free sugars ≥ 10% of EER (PR 1.42, 95% CI 1.42–1.91). Children from households with the greatest socioeconomic disadvantage were more likely to be in the top tertile for free sugars intake (PR 1.58, 95% CI 1.19–2.10) and free sugars ≥ 10% of EER (PR 1.44, 95% CI 1.13–1.84) than the least disadvantaged. Children of mothers with some university level education were less likely to have a free sugars intake ≥ 10% of EER (PR 0.82, 95% CI 0.69–0.98), while the association with being in the top tertile of free sugars intake just failed to reach significance (PR 0.83, 95% CI 0.68–1.01).

### 3.6. Sensitivity Analysis

Removal of the 125 participants who were assigned a reference EER due to missing weight data resulted in similar overall findings (Appendix A). In this subset, the median free sugars intake was 22.0 (IQR 12.7–37.6) g/day, with 70.2% of children exceeding the WHO 5% recommendation, and 37.4% exceeding the 10% recommendation.

## 4. Discussion

To our knowledge, this is the first study to investigate free sugars intakes of Australian children from 1 to 2 years of age. Our findings suggest that for many children, recommendations to limit intakes of foods high in added or free sugars during the early years are not being met. Free sugars intakes appear to have increased from 1 to 2 years of age, both in terms of absolute intakes and as a percentage of energy. At 2 years of age, most of the cohort had free sugars intakes that exceeded the WHO 5% recommendation, and almost two fifths exceeded the 10% recommendation. We also found that exceeding the 10% recommendation and being in the highest tertile for free sugars intakes was increasingly observed as socioeconomic disadvantage increased.

The 1 and 2 year free sugars data reported here show that the increase in free sugars intake observed from 2 years on in the National Nutrition and Physical Activity Survey (NNPAS), apparently begins early in life. Similar to the findings for older children in the NNPAS [8], we observed an increase in usual daily intakes of free sugars with age, from median intake of 7.3 g at 1 year of age [22] to 22.5 g in the same cohort at 2 years of age. These findings are lower than the median free sugars intakes at 2–3 years in the NNPAS, which may be due to the slightly older participants in the national survey, different dietary assessment methods, an increased interest in avoiding sugar in popular media influencing our more recent cohort [37,38], social desirability bias in our oral health study affecting sugars reporting, or other factors that remain unknown. 

Free sugars intakes as a proportion of energy appear to have increased from 1 to 2 years of age, however these measures differed between the two time points and so these findings should be interpreted cautiously. At 1 year of age, usual energy intake was measured by dietary assessment methods, whereas at 2 years an estimated energy requirement was calculated for each child from their measured weight. Nevertheless, there appears to be a large increase in proportion of energy coming from free sugars, from a mean of 3.6 ± 2.8% of energy intake at 1 year [22] to 10.4 ± 8.6% of EER at 2 years. Similarly, those exceeding the WHO < 10% recommendation increased from 2.4% of the cohort at 1 year to 38.4% of the cohort at 2 years; and the <5% recommendation was exceeded by 71.1% at 2 years, up from 22.8% at 1 year [22]. Even with reticence toward the specific values reported here, an overall upward trend is clear.

Similar to our findings at 1 year of age, free sugars intakes at 2 years followed a socioeconomic gradient, with increased likelihood of exceeding the WHO recommendation, and of being in the top tertile for free sugars intake among those in lower IRSAD quintiles. This finding is reflected elsewhere among children and adults [19,20,39,40], and highlights the importance of addressing social determinants across the life course in future efforts to improve dietary behaviors. 

High free sugars intakes were more common among children whose mother was younger and less educated. These findings are consistent with studies that have investigated determinants of diet quality in early childhood, with children of younger mothers and less educated mothers being consistently reported to have poorer quality diets than children of older and more educated women [19,20,21,41,42,43]. Studies have found that the diets of mothers and young children are correlated [42,44] and that education level is associated with diet quality in adults [45], which likely explains the association between maternal education and child diet quality. Additionally, associations have been found between lower maternal education and the use of food as reward for child behavior [46], which usually includes non-core foods.

The prevalence of being a high free sugars consumer was also greater for children from larger families, and for those whose mother was not born in Australia or New Zealand. It has been suggested that demand from older siblings may increase household exposure to non-core foods, and that caregiving responsibilities of parents may be stretched further in households with multiple children, limiting time for meal preparation [10,41]. There are many possible contributing factors to why a mother not born in Australia or New Zealand may be a determinant of high free sugars consumption. Two thirds of SMILE mothers born outside of Australia were born in Asian countries, where there may be cultural norms around sweetening particular foods. For example, providing sweetened tea or milk drinks from a young age is common practice in some Asian and African cuisines [47,48,49]. Families undergoing nutrition transition tend to retain traditional staple foods, but incorporate the accessory foods of the new country, which are usually non-core foods [50]. Different expectations around the role and autonomy of children in the household may give children a greater degree of influence on household food intake [51]. Overall, dietary acculturation has generally resulted in a decline in diet quality as families move to a more Western dietary pattern high in non-core foods and associated with affluence [50,52].

The major sources of free sugars in our cohort at 2 years were mostly non-core foods, and were similar to the major food sources reported in the NNPAS for older children. Fruit and vegetable juices and drinks provided the greatest overall contribution to free sugars intakes in both the NNPAS (25% of free sugars) at 2–3 years and our cohort (11%) at 2 years, and were consumed by 44% of the 2–3 year NNPAS responders and 57% of our cohort. In comparison, approximately 12% of our participants consumed this food group at 1 year of age, giving a lower overall contribution to free sugars (7.4%). So from 1 to 2 years of age, we observed an increase in the number of consumers, mean grams consumed and overall percentage contribution of free sugars coming from fruit and vegetable juices and drinks. Our finding reflects those of Byrne, et al. [12] in another Australian cohort that observed a similar increase in fruit juice-based drinks coupled with a decline in milk-based drinks from 2 to 5 years of age. As the WHO definition of free sugars includes sugars naturally present in fruit juices and concentrates, fruit and vegetable juices and drinks are recognized as a sugar sweetened beverage, with a contributing role in obesity and dental decay [5]. As such, recommendations to limit intakes of fruit juice in children of all ages should be prioritized [4,53].

Infant and toddler foods are a point of difference between our cohort and the NNPAS. These foods appear to have a decreasing role in the diets of toddlers, with fewer consumers and lower overall contribution to free sugars from 1 to 2 years. However, in our cohort this food group still contributed approximately 15% of free sugars at 2 years of age, down from 28% at 1 year, but much greater than the 0.1% reported in the NNPAS. Possible reasons for this discrepancy include more steps taken to differentiate these foods in our data collection methods at both time points, limitations of the infant foods categories in AUSNUT2011–13 and the substantial growth of the infant and toddler food market in recent years [22,54]. Future updates to national nutrient databases will need substantial revisions to the food list and infant food group categories, as the AUSNUT2011–13 database does not reflect the current infant and toddler food market.

Yogurt was the only core food group contributing substantially to free sugars intakes among all children, with higher proportion in our cohort at 1 year (9.6%) and 2 years (8.8%) than the 6.2% reported in the NNPAS for 2–3 year olds. We also recorded an additional 9.8% of free sugars from the minor food group infant and toddler custards or yogurts at 2 years, which, although not directly reported in the NNPAS, contributed less than 0.1% of free sugars as part of the sub-major food group infant foods. In addition to the already described reasons for these discrepancies, yogurts have particularly high variation in sugars content, making this food group difficult to capture accurately in dietary surveys. Products range from plain, unsweetened yogurt with zero free sugars, to flavored yogurts containing over 15 g of free sugars per 100 g. Children’s yogurt pouches seem to exemplify this variation, with products appearing side-by-side on supermarket shelves that contain a three-to-four-fold difference in sugar content for the same flavor across different brands. As a core food, yogurt is an ideal early food for infants, but products with little or no free sugars should be selected. This finding highlights the importance of transparent food labelling that includes free sugars, consumer education and mandatory front-of-pack nutrition labelling to support consumers and encourage product reformulation [53,55,56].

At present, the Australia New Zealand Food Standards code does not sufficiently regulate the use of free sugars in infant and toddler food products [22]. Food manufacturers avoid the standard for infant products by labelling foods as suitable for children over 12 months of age. They are also able to sweeten infant foods using fruit juice or fruit juice concentrate, in order to avoid the labelling requirement that infant foods containing more than 4 g/100 g added sugars must be labelled as “sweetened” [57]. Definitions of added sugars often differ from that of free sugars by the exclusion of fruit juice, which has left a loophole in the current food standard. Given the rapid growth in the infant and toddler food market in recent years, the current standards have fallen behind, and action is needed by policymakers to ensure foods marketed to infants and toddlers are low in free sugars, and clearly labelled.

As with compliance with WHO recommendations, direct comparisons of food group contributions should be interpreted cautiously. Slightly different methods were used to assign food group contributions at the two time points, and food groups in the 2 year data were retrospectively assigned to the SMILE-FFQ, which was designed with a focus on sugars at a nutrient rather than food group level. However the AUSNUT2011–13 food groups were used as consistently as possible, with most line items in the SMILE-FFQ matching directly to one major or sub-major food group.

A notable point of difference in the nutrient output between the SMILE-FFQ and the AUSNUT2011–13 database were the attribution of a free sugars value to infant and toddler formula. This was assigned a zero value for free sugars in AUSNUT2011–13, however some of these products, particularly those made from plant-based milk alternatives such as soy, and to a lesser extent toddler formula, contain added sucrose and other sources of free sugars. When designing the nutrient database for the SMILE-FFQ we assigned a conservative free sugars value to the line item for formula, based on cow’s milk-based toddler formula, following the method for determining free sugars adapted from Louie, et al. [27]. The result was higher than the zero value from most cow’s-milk based infant formula, but lower than formula from plant-based alternatives. Similar to other dairy milk alternatives, formula alternatives may be a key source of free sugars among those who consume them. It was not possible to separate out these different formula types due to the single line item in the SMILE-FFQ, and the relatively crude infant food group categories of AUSNUT2011–13. As a result, the contribution to free sugars among consumers of plant-based formula is likely to be underestimated, with a corresponding overestimation for consumers of animal milk-based infant formula.

Given the limited reporting of dietary intakes in the first 2 years of life and the recent release of the WHO guidelines for sugars consumption among adults and children [5], this research provides valuable insight into the early establishment of free sugars intakes of Australian children. We support the recent recommendation by Spence, et al. [19], that national dietary surveys should include children younger than 2 years of age in order to obtain a nationally representative and consistent picture of the establishment of dietary patterns throughout early childhood. However, more work is needed to ensure that infant and toddler foods are accurately captured, via modifications to survey design method and revisions to the food list and infant food groups in the national nutrient database. It is likely that discrepancies between our findings and the most recent NNPAS can be attributed to these methodological and age differences.

Strengths of this study include our cohort recruitment and dietary assessment methods. Initial efforts to over-sample from hospitals in socially disadvantaged areas to address attrition bias have resulted in a study population that is socioeconomically diverse and generally representative of the population, based on South Australian perinatal statistics collected in 2013 [24].

The use of a customized FFQ that was designed to capture free sugars in Australian toddlers and validated with an external cohort prior to administration provides robust and valid free sugars intake data at 2 years of age [26]. The use of a 24-h recall and 2-day food diary on non-consecutive days at 1 year of age also resulted in a reliable measure of usual free sugars intake, tailored to the infant and toddler food landscape. Due to the different dietary assessment methods at the two time points, statistical comparisons focused on movement relative to the cohort, rather than absolute values. 

A limitation of the study is the relatively small proportion of cases (10%) that were lost from the determinant analysis due to incomplete data for one or more sociodemographic characteristics. However, previous sensitivity analysis for other dietary outcomes investigated in this cohort, in which missing data was imputed under the assumption that data were missing at random, revealed that distributions of variables in the imputed data sets were consistent with the complete case data set [58]. 

All dietary assessment methods have inherent limitations in precision and accuracy, including social desirability bias and other forms of misreporting [31,59,60]. In our case, proxy reporting was used due to the age of the cohort, and so the reports may not reflect food intakes of the child when in the care of others. Proxy estimates of quantity intakes of toddlers are likely to be more difficult than of older children due to the small and variable portion sizes experienced at this age, greater frequency of meals and snacks and higher plate waste [61,62]. 

The association between sugars intake and dental caries is well-known, and our cohort are aware that SMILE is an oral health study. Participants are likely to have underestimated free sugars intake due to social desirability bias, particularly at 2 years when they were due to have their oral examination. Some of this underestimation may be ameliorated by the use of EER for determining compliance with the WHO recommendations. In the NNPAS mean daily energy intakes of 2–3 year olds exceeded the EER [40], so it is likely that using EER with true free sugars intakes would underestimate actual energy intake and falsely elevate the percentage of energy from free sugars. It is not possible to determine the extent of this bias, or whether it is consistent across the cohort. Underestimation of free sugars or EER may be greater in sub-sets of the cohort, for example those at the extremes of weight values [63], those whose actual energy intake is substantially over or under the EER, or those for whom EER was imputed in the absence of weight data.

## 5. Conclusions

This study reports free sugars intakes and food sources among 1 and 2 year old children; an age group not currently included in national dietary monitoring. Findings suggest that free sugars enter the diet somewhat conservatively in the first year of life, and then increase substantially by 2 years of age. National data in older children supports this perceived trend. Noncompliance with the WHO recommendations for free sugars intakes has also increased at 2 years of age, with most of the cohort exceeding the 5% recommendation, and almost two fifths exceeding the 10% recommendation. For the most part, food sources of free sugars reflect those of older children. These are predominantly non-core foods, such as fruit juice, biscuits, cakes, desserts and confectionery; with yogurt and non-dairy milk alternatives the two notable exceptions. As observed in other settings, we found that diet quality follows a social gradient, with high free sugars consumption more prevalent among 2 year olds from households with greater socioeconomic disadvantage.

These findings highlight the need for improved efforts to prevent provision of sources of free sugars in the first years of life. It is important to establish family food and drink intakes that are centered on core foods, and set-up lifelong eating patterns that are associated with reduced risk of obesity, dental caries and other chronic disease. Policymakers and practitioners should consider approaches that support parents in food selection and encourage product reformulation by manufacturers, such as improving the food standard for infant foods to include toddler products and incorporate the WHO definition of free sugars, mandatory front-of-pack nutrition labelling and the addition of free sugars to the nutrition information panel for all products. Parents should be encouraged and supported to limit early exposure to foods high in free sugars, particularly fruit juice and non-core foods. Strategies to address social determinants of health and reduce inequality must be considered, and future interventions should target groups with greater socioeconomic disadvantage. Progress in these efforts should be evaluated via ongoing national dietary surveys, to monitor intakes of free sugars and non-core foods throughout the course of life.

## Figures and Tables

**Figure 1 nutrients-11-00161-f001:**
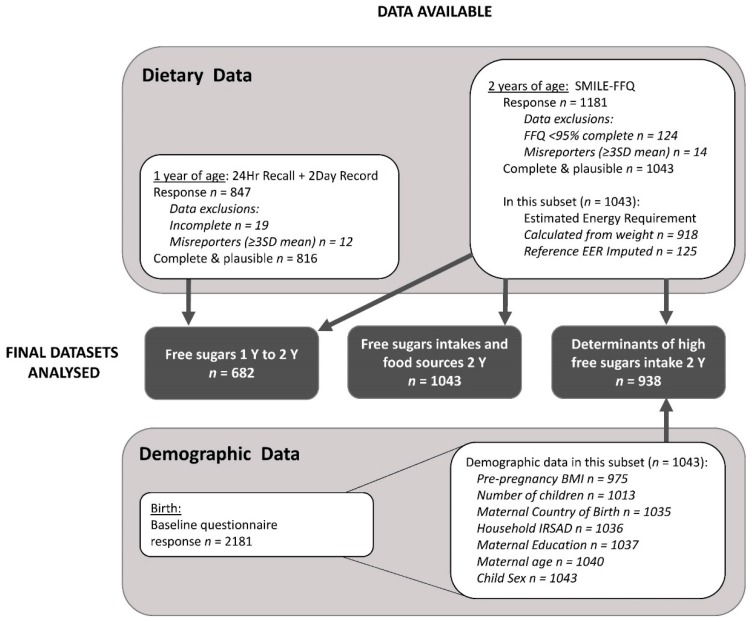
Participant flowchart for the analysis datasets. SMILE-FFQ: Study of Mothers’ and Infants’ Life Events Food Frequency Questionnaire; SD: Standard Deviation; Y: year; BMI: Body Mass Index IRSAD: Index of Relative Socioeconomic Advantage and Disadvantage.

**Table 1 nutrients-11-00161-t001:** Participant characteristics.

	All Participants (*n* = 2181)			Responders 2 Years (*n* = 1043)			Non-Responders 2 Years (*n* = 1138)			*p ^a^*
	*n*	mean	SD	*n*	mean	SD	*n*	mean	SD	
Mother’s age at birth (years)	2101	29.9	5.4	1040	30.6	5.0	1061	29.2	5.7	<0.001
Total number of children	1867	1.8	1.0	1013	1.8	0.9	854	1.9	1.0	0.012
Child’s birthweight (g)	2083	3356	572	1032	3393	547	1051	3319	593	0.003
	***n***	**%**		***n***	**%**		***n***	**%**		
Mother’s age at birth (years)										<0.001
<25	336	16.0		105	10.1		231	21.8		
25–34.99	1353	64.4		717	68.9		636	59.9		
≥35	412	19.6		218	21.0		194	18.3		
Mother’s pre-pregnancy BMI (kg/m^2^)										0.274
<25	1086	56.2		565	57.9		521	54.4		
25–29.99 (overweight)	455	23.5		218	22.4		237	24.7		
≥30 (obese)	392	20.3		192	19.7		200	20.9		
Mother’s Country of Birth										<0.001
Australia or New Zealand	1453	69.3		755	72.9		698	65.7		
Other	644	30.7		280	27.1		364	34.3		
Mother’s Education										<0.001
high school/vocational	1136	54.0		451	43.5		685	64.2		
some university and above	968	46.0		586	56.5		382	35.8		
IRSAD										<0.001
Deciles 1–2 (most disadvantaged)	462	22.2		172	16.6		290	27.8		
Deciles 3–4	446	21.4		214	20.7		232	22.2		
Deciles 5–6	390	18.7		210	20.3		180	17.2		
Deciles 7–8	385	18.5		199	19.2		186	17.8		
Deciles 9–10 (most advantaged)	398	19.1		241	23.3		157	15.0		
Total number of children										0.111
1	863	46.2		480	47.4		383	44.8		
2	670	35.9		369	36.4		301	35.2		
≥3	334	17.9		164	16.2		170	19.9		
Child’s sex										0.579
Male	1146	52.7		556	53.3		590	52.1		
Female	1029	47.3		487	46.7		542	47.9		

^a^*p*-values for responders v non-responders at 2 years using *t*-test (continuous) or Pearson Chi Square test (categorical). BMI: Body Mass Index; IRSAD: Index of Relative Socioeconomic Advantage and Disadvantage.

**Table 2 nutrients-11-00161-t002:** Free Sugars intakes at 2 years of age (*n* = 1043).

	Median	Percentile		Range
		25th	75th	
Free Sugars (g/day)	22.5	12.8	37.7	0.3–140.7
Tertile 1 (low)	10.6	7.4	13.0	0.3–15.8
Tertile 2 (mid)	22.7	19.0	26.4	15.81–31.6
Tertile 3 (high)	47.2	38.0	66.7	31.61–140.7
Total Sugars (g/day)	77.5	56.0	105.3	9.4–294.8
Estimated Energy Requirement (kJ/day)	4730	4357	5214	2644–8193
Percentage of Estimated Energy Requirement from Free Sugars (%)	8.0	4.6	13.2	0.1–61.3

**Table 3 nutrients-11-00161-t003:** Food group contributions to free sugars intakes at 2 years of age.

	All Participants (*n* = 1043)				Consumers				
Food group ^a^	mean ± SD free sugars (g)			% contribution to free sugars	*n* (%)		mean ± SD free sugars (g)		
Non-alcoholic beverages	4.7	±	9.7	15.9	1037	(99.4)	4.7	±	9.7
Fruit and vegetable juices, and drinks	3.1	±	7.3	10.7	590	(56.6)	5.5	±	9.0
Cordials	0.8	±	3.8	2.6	179	(17.2)	4.5	±	8.4
Soft drinks, flavored mineral waters	0.2	±	1.3	0.8	147	(14.1)	1.7	±	3.1
Other beverage flavorings	0.3	±	1.7	0.9	246	(23.6)	1.2	±	3.5
Cereals and cereal products	1.1	±	2.1	3.7	953	(91.4)	1.2	±	2.1
Breakfast cereals, ready to eat	1.1	±	2.1	3.7	884	(84.8)	1.3	±	2.2
Cereal based products and dishes	5.5	±	6.9	18.8	1028	(98.6)	5.6	±	6.9
Sweet biscuits	2.2	±	3.5	7.4	908	(87.1)	2.5	±	3.6
Savory biscuits	0.3	±	0.5	0.9	947	(90.8)	0.3	±	0.5
Cakes, muffins, doughnuts, cake-type desserts ^b^	2.2	±	4.2	7.5	674	(64.6)	3.4	±	4.8
Batter-based products, scones, sweet breads ^b^	0.8	±	1.6	2.6	468	(44.9)	1.7	±	2.0
Fruit products and dishes ^c^	0.3	±	1.2	1.2	1029	(98.7)	0.3	±	1.2
Milk products and dishes	4.8	±	7.1	16.4	1015	(97.3)	4.9	±	7.2
Yogurt	2.6	±	5.2	8.8	787	(75.5)	3.4	±	5.8
Frozen milk products	0.9	±	1.9	3.1	605	(58.0)	1.5	±	2.2
Custards	0.9	±	3.3	2.9	325	(31.2)	2.8	±	5.5
Flavored milks and milkshakes	0.4	±	1.5	1.2	211	(20.2)	1.8	±	3.0
Dairy and meat substitutes	1.7	±	4.9	5.8	438	(42.0)	4.0	±	6.9
Dairy milk substitutes, unflavored	1.2	±	3.8	4.2	245	(23.5)	5.2	±	6.5
Savory sauces and condiments	0.5	±	1.0	1.8	792	(75.9)	0.7	±	1.2
Gravies and savory sauces	0.5	±	0.9	1.6	758	(72.7)	0.6	±	1.0
Sugar products and dishes	3.4	±	5.7	11.7	846	(81.1)	4.2	±	6.0
Sugar, honey and syrups	1.5	±	3.5	5.1	609	(58.4)	2.6	±	4.3
Jam and lemon spreads, chocolate spreads, sauces	1.0	±	2.7	3.3	541	(51.9)	1.8	±	3.5
Dishes and products other than confectionery where sugar is the major component	1.0	±	2.8	3.3	450	(43.1)	2.2	±	3.9
Confectionery and cereal, nut, fruit, seed bars	2.8	±	3.9	9.4	871	(83.5)	3.3	±	4.0
Chocolate and chocolate-based confectionery	1.3	±	2.5	4.6	719	(68.9)	1.9	±	2.8
Muesli or cereal style bars	0.7	±	1.6	2.4	404	(38.7)	1.8	±	2.2
Other confectionery	0.7	±	1.7	2.5	439	(42.1)	1.7	±	2.4
Infant formula and foods	4.3	±	5.9	14.9	770	(73.8)	5.9	±	6.2
Infant and Toddler formula	1.1	±	3.3	3.8	150	(14.4)	7.7	±	5.1
Infant foods	2.8	±	4.4	9.7	658	(63.1)	4.5	±	4.8
Infant custards or yogurts	2.8	±	4.4	9.7	658	(63.1)	4.5	±	4.8
Infant drinks ^d^	0.4	±	1.7	1.3	155	(14.9)	2.7	±	3.8
Infant fruit juices	0.4	±	1.7	1.3	155	(14.9)	2.7	±	3.8

^a^ Food groups providing < 1% contribution to free sugars not listed. ^b^ Group variation from AUSNUT to match FFQ line items, no difference at major food group level. ^c^ Free sugars in fruit products and dishes come only from canned fruit in syrup and stewed fruit with added sugar. ^d^ Infant drinks category includes infant fruit juices, but does not include infant formula or human breast milk.

**Table 4 nutrients-11-00161-t004:** Tertile of free sugars intakes from 1 to 2 years of age (*n* = 682).

		Free Sugars at 2 Years		
		Low	Mid	High
**Free Sugars at 1 Year**	**Low**	118 (17.3%)	72 (10.6%)	37 (5.4%)
	**Mid**	82 (12.0%)	76 (11.1%)	70 (10.3%)
	**High**	49 (7.2%)	67 (9.8%)	111 (16.3%)

Weighted Kappa 0.241, *p* < 0.001.

**Table 5 nutrients-11-00161-t005:** Participant characteristics associated with high consumption of free sugars at 2 years (*n* = 938).

	Free Sugars Highest Tertile		Free Sugars Intake ≥ 10%EER	
	PR	95% CI	PR	95% CI
Mother’s age at birth (years)				
<25	1.50	1.07–2.12	1.42	1.05–1.91
25–34.99	1.06	0.84–1.34	1.05	0.85–1.29
≥35	1.00		1.00	
Mother’s pre-pregnancy BMI				
<25 Healthy weight or below	1.00		1.00	
25–29.99 Overweight	1.06	0.85–1.32	1.03	0.85–1.24
≥30 Obese	0.95	0.74–1.23	0.92	0.73–1.17
Mother’s Country of Birth				
Other	1.58	1.28–1.94	1.59	1.33–1.90
Australia, New Zealand	1.00		1.00	
Mother’s Education				
University	0.83	0.68–1.01	0.82	0.69–0.98
High school/vocational	1.00		1.00	
IRSAD				
Deciles 1–2 (most disadvantaged)	1.58	1.19–2.10	1.44	1.13–1.84
Deciles 3–4	1.40	1.06–1.85	1.34	1.06–1.70
Deciles 5–6	1.03	0.76–1.40	0.98	0.75–1.29
Deciles 7–8	1.00	0.73–1.38	0.95	0.72–1.25
Deciles 9–10 (most advantaged)	1.00		1.00	
Total number of children				
1	1.00		1.00	
2	1.28	1.04–1.57	1.22	1.02–1.47
≥3	1.52	1.17–1.97	1.39	1.11–1.76
Child’s sex				
Female	0.93	0.77–1.11	0.98	0.83–1.15
Male	1.00		1.00	

PR: Prevalence Ratio; 95% CI: 95% Confidence Interval; BMI: Body Mass Index (kg/m^2^); IRSAD: Index of Relative Socioeconomic Advantage and Disadvantage.

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
