# Peer review of "Free Sugars Intake, Sources and Determinants of High Consumption among Australian 2-Year-Olds in the SMILE Cohort"

_nutrients, 2019, doi:10.3390/nu11010161_

Round 1

Reviewer 1 Report

I found this article, on an understudied topic to be interesting and well written. I have a few comments and suggestions. 

Introduction:

Please define "free sugar". The term is qualified in line 297 of the discussion, but a more clear definition earlier in the article would be useful for readers. 

Materials and Methods:

Please provide justification as to why the 125 participants with no weight measurements were assigned a reference EER rather than being excluded from the sample. If the participants had not been included would the findings of the study have differed?

Table 3:

the n and % labels under consumers should be combined into one column n(%). Also, the % contribution to free sugars data should be included under the consumers section as it is for non-consumers. 

Author Response

Thank you for your comments.

Point 1. Introduction: Please define "free sugar". The term is qualified in line 297 of the discussion, but a more clear definition earlier in the article would be useful for readers.

Response 1: The WHO definition of free sugars has been added to the introduction (line 40)

Point 2. Materials and Methods: Please provide justification as to why the 125 participants with no weight measurements were assigned a reference EER rather than being excluded from the sample. If the participants had not been included would the findings of the study have differed?

Response 2: The 125 participants without a weight measurement were assigned as reference EER in order to maximise the sample size.  Sensitivity analysis performed with the 125 removed and provided as supplementary materials. This has been added to the methods (line 111) and results (line 251 on).

Point 3. Table 3: the n and % labels under consumers should be combined into one column n(%). Also, the % contribution to free sugars data should be included under the consumers section as it is for non-consumers.

Response 3: Table 3 changes made as suggested.

Reviewer 2 Report

This is well-written and interesting paper reporting free sugars intake and determinants of high consumption among an under-studied population. Some major and minor comments as follows:
Major:
Statistical method, how did you determine the final multivariable model? it is not appropriate to just include all explanatory variables in the model. could relative risk be estimated as it is much easier to interpret? how did you deal with missing data/lost to follow-up?
Figure 1 is a bit difficult to follow. could it be better organised?
it would be interesting to see also the determinants of the change in free sugars intakes between 1 and 2 years of age. movements between tertiles could be combined into positive or negative change or the change in standardised intakes between two time points could be used as a continuous variable to indicate the relative change (impact of different method could be diminished)
Minor:
where are the non-responders data in table 1?
any fathers’ information, such as education level and BMI?
why were two methods used to compare the change in tertiles between 1 and 2 years of age?
please remove p values from table 5 as 95% CI was presented.

Author Response

Thank you for your comments.

Point 1: Statistical method, how did you determine the final multivariable model? it is not appropriate to just include all explanatory variables in the model. 

Response 1: Variables included in the final model were those identified in the literature as being associated with poor diet quality in children. Household income was available but not used due to similarities with the Index of relative Social Advantage and Disadvantage (IRSAD) which is a composite measure of socio-economic position, and not wanting to over-adjust. Due to small number of explanatory variables relative to the large sample size, all potential explanatory variables were entered into the multivariable model, as instructed by our statistical consultant. We have clarified this in the methods section (line 161)

Point 2: could relative risk be estimated as it is much easier to interpret? 

Response 2: The analysis of the 2 year data was cross-sectional and therefore it is not appropriate to calculate relative risk

Point 3: how did you deal with missing data/lost to follow-up?

Response 3: Participants with data missing for one or more of the explanatory variables were not included in the analysis population, as described in the original submission (line 165)

Point 4: Figure 1 is a bit difficult to follow. could it be better organised?

Response 4: The figure has been revised, but as this was not identified as being a problem by the other two reviewers we do not think that major changes are warranted.

Point 5: it would be interesting to see also the determinants of the change in free sugars intakes between 1 and 2 years of age. movements between tertiles could be combined into positive or negative change or the change in standardised intakes between two time points could be used as a continuous variable to indicate the relative change (impact of different method could be diminished)

Response 5: Because of the different methods used to collect free sugars and to estimate the contribution of free sugars to total energy intake at the two time points, we took a conservative approach to this analysis. While we think that the analysis suggested by the reviewer would indeed be interesting, we do not think that the data are robust enough to warrant this more detailed analysis.

Point 6: where are the non-responders data in table 1?

Response 6: Non responders are different at 1 year and 2 years, and the inclusion of these in the original version of Table 1 would make for a complicated table. The requested data are presented via a new version of Table 1 which provides the non-responders with incomplete 2 year questionnaire, which is the source of data for the primary outcomes investigated in this analysis; and a supplementary table which provides the non-responder data for those who did not have complete year 1 and year 2 data. 

Point 7: any fathers’ information, such as education level and BMI?

Response 7: These were not used in the model as maternal education and BMI were identified in the literature as the key explanatory variables. 

Point 8: why were two methods used to compare the change in tertiles between 1 and 2 years of age?

Response 8: Table 4 and results section 3.3 changed to only include Weighted Kappa

Point 9: please remove p values from table 5 as 95% CI was presented

Response 9: p values removed

Reviewer 3 Report

This is a thorough and well-written paper from a highly experienced team which presents new information about the free sugar intakes of very young children. It is an important finding which will allow the development of education programs, interventions and policies to help parents and support the health of young children.

I felt there was only one omission in the discussion and that was putting the onus back onto the well-funded food manufacturers and FSANZ to produce or mandate processed food production for infants and young children that has low sugar content rather than the parents.

Author Response

Point: I felt there was only one omission in the discussion and that was putting the onus back onto the well-funded food manufacturers and FSANZ to produce or mandate processed food production for infants and young children that has low sugar content rather than the parents.

Response: Thank you for this feedback. We have added a paragraph in the discussion (line 391 onwards), discussing the limitations of the current food standard for infants. We have also made revisions to the final paragraph recommendations for policymakers (line 463), and swapped the order in that paragraph to emphasise industry/policymaker action.

Round 2

Reviewer 2 Report

Two issues must be addressed before this manuscript can be accepted for publication.

RR could be calculated (i.e., Prevalence ratio for cross-sectional design, see a paper https://www.ncbi.nlm.nih.gov/pmc/articles/PMC5135596/). the author should do so or provide a convincing explanation why OR outperforms RR (i.e., PR). 

complete case analysis does not account for missingness and its impact on generalizability should be discussed.

Author Response

Thank you for your comments.

Point 1: RR could be calculated (i.e., Prevalence ratio for cross-sectional design, see a paper https://www.ncbi.nlm.nih.gov/pmc/articles/PMC5135596/). the author should do so or provide a convincing explanation why OR outperforms RR (i.e., PR).

Response 1: Odds ratios changed to prevalence ratios. New version of table 5 and supplementary table S5 provided. Most findings did not differ, however maternal age and level of education became significant. Revisions made to methods (line 162 on), results (line 218 on) and discussion (line 275 on) to reflect this change.

Point 2: complete case analysis does not account for missingness and its impact on generalizability should be discussed.

Response 2: A sensitivity analysis was conducted in an earlier analysis involving the SMILE cohort to explore the impact of missing data on generalisability. 

Ha, D., L. Do, A. Spencer, W. Thomson, R. Golley, A. Rugg-Gunn, S. Levy and J. Scott (2017). "Factors Influencing Early Feeding of Foods and Drinks Containing Free Sugars—A Birth Cohort Study." International Journal of Environmental Research and Public Health 14(10): 1270. 

To minimise bias, missing data on exposure variables were imputed under the assumption that data were missing at random. Twenty imputed data sets were generated using SAS Proc MI and the results of the imputed analyses were combined. The distributions of variables in the imputed data sets were consistent with the complete case data. 

We have acknowledged in the discussion that missing data were a limitation of the current study and refer to the previously conducted sensitivity analysis to confirm the credibility of the analysis (line 390 on).